# On Classical Gauss Sums and Some of Their Properties

**Li Chen**

School of Mathematics, Northwest University, Xi'an 710127, China; cl1228@stumail.nwu.edu.cn

**Abstract:** The goal of this paper is to solve the computational problem of one kind rational polynomials of classical Gauss sums, applying the analytic means and the properties of the character sums. Finally, we will calculate a meaningful recursive formula for it.

**Keywords:** third-order character; classical Gauss sums; rational polynomials; analytic method; recursive formula

## 1. Introduction

Let $q \geq 3$ be an integer. For any Dirichlet character $\chi$ mod $q$, according to the definition of classical Gauss sums $\tau(\chi)$, we can write

$$\tau(\chi) = \sum_{a=1}^{q} \chi(a) e\left(\frac{a}{q}\right),$$

where $e(y) = e^{2\pi i y}$.

Since this sum appears in numerous classical number theory problems, and it has a close connection with the trigonometric sums, we believe that classical Gauss sums play a crucial part in analytic number theory. Because of this phenomenon, plenty of experts have researched Gauss sums. Meanwhile, more conclusions have been obtained as regards their arithmetic properties. Such as the following results provided by Chen and Zhang [1]:

Let $p$ be an odd prime with $p \equiv 1 \bmod 4$, $\lambda$ be any fourth-order character mod $p$. Then one has the identity

$$\tau^2(\lambda) + \tau^2\left(\overline{\lambda}\right) = \sqrt{p} \cdot \sum_{a=1}^{p-1}\left(\frac{a+\overline{a}}{p}\right) = 2\sqrt{p} \cdot \alpha,$$

where $\left(\frac{*}{p}\right) = \chi_2$ denotes the the Legendre's symbol mod $p$ (please see Reference [1,2] for its definition and related properties), and $\alpha = \sum_{a=1}^{\frac{p-1}{2}}\left(\frac{a+\overline{a}}{p}\right)$.

If $p$ is a prime with $p \equiv 1 \bmod 3$, $\psi$ is any third-order character mod $p$, then Zhang and Hu [3] had already obtained an analogous result (see Lemma 1). However, perhaps the most beautiful and important property of Gauss sums $\tau(\chi)$ is that $|\tau(\chi)| = \sqrt{q}$, for any primitive character $\chi$ mod $q$.

Reference [2] and References [4–13] have a good deal of various elementary properties of Gauss sums. In this paper, the following rational polynomials of Gauss sums attract our attention.

$$U_k(p,\chi) = \frac{\tau^{3k}(\chi)}{\tau^{3k}(\overline{\chi})} + \frac{\tau^{3k}(\overline{\chi})}{\tau^{3k}(\chi)}, \tag{1}$$

where $p$ is an odd prime, $k$ is a non-negative integer, $\chi$ is any non-principal character mod $p$.

Observing the basic properties of Equation (1), we noticed that hardly anyone had published research in any academic papers to date. We consider that the question is significant. In addition, the regularity of the value distribution of classical Gauss sums could be better revealed. Presently, we will explain certain properties discovered in our investigation. See that $U_k(p,\chi)$ has some good properties. In fact, for some special character $\chi$ mod $p$, the second-order linear recurrence formula for $U_k(p,\chi)$ for all integers $k \geq 0$ may be found similarly.

The goal of this paper is to use the analytic method and the properties of the character sums to solve the computational problem of $U_k(p,\chi)$, and to calculate two recursive formulae, which are listed hereafter:

**Theorem 1.** *Let $p$ be a prime with $p \equiv 1$ mod 12, $\psi$ be any third-order character mod $p$. Then, for any positive integer k, we can deduce the following second-order linear recursive formulae*

$$U_{k+1}(p,\psi) = \frac{d^2 - 2p}{p} \cdot U_k(p,\psi) - U_{k-1}(p,\psi),$$

*where the initial values $U_0(p,\psi) = 2$ and $U_1(p,\psi) = \frac{d^2-2p}{p}$, d is uniquely determined by $4p = d^2 + 27b^2$ and $d \equiv 1$ mod 3.*

So we can deduce the general term

$$U_k(p,\psi) = \left( \frac{d^2 - 2p + 3dbi\sqrt{3}}{2p} \right)^k + \left( \frac{d^2 - 2p - 3dbi\sqrt{3}}{2p} \right)^k, \quad i^2 = -1.$$

**Theorem 2.** *Let $p$ be a prime with $p \equiv 7$ mod 12, $\psi$ be any third-order character mod $p$. Then, for any positive integer k, we will obtain the second-order linear recursive formulae*

$$U_{k+1}(p,\psi) = \frac{i(2p - d^2)}{p} \cdot U_k(p,\psi) - U_{k-1}(p,\psi),$$

*where the initial values $U_0(p,\psi) = 2$, $U_1(p,\psi) = \frac{i(2p-d^2)}{p}$ and $i^2 = -1$.*

Similarly, we can also deduce the general term

$$U_k(p,\psi) = i^k \left( \frac{2p - d^2 + \sqrt{8p^2 - 4pd^2 + d^4}}{2p} \right)^k + i^k \left( \frac{2p - d^2 - \sqrt{8p^2 - 4pd^2 + d^4}}{2p} \right)^k.$$

## 2. Several Lemmas

We have used five simple and necessary lemmas to prove our theorems. Hereafter, we will apply relevant properties of classical Gauss sums and the third-order character mod $p$, all of which can be found in books concerning elementary and analytic number theory, such as in References [2,10], so we will not duplicate the related contents.

**Lemma 1.** *If $p$ is any prime with $p \equiv 1 \bmod 3$, $\psi$ is any third-order character* mod $p$, *then, we have the equation*

$$\tau^3(\psi) + \tau^3(\overline{\psi}) = dp,$$

*where $\tau(\psi)$ denotes the classical Gauss sums, $d$ is uniquely determined by $4p = d^2 + 27b^2$ and $d \equiv 1 \bmod 3$.*

**Proof.** See References [3] or [8]. □

**Lemma 2.** *Let $p$ be a prime with $p \equiv 1 \bmod 3$, $\psi$ be any third-order character* mod $p$, $\chi_2 = \left(\frac{*}{p}\right)$ *denotes the Legendre's symbol* mod $p$. *The following identity holds*

$$\tau^2(\overline{\psi}) = \left(\frac{-1}{p}\right)\psi(4)\tau(\chi_2)\tau(\psi\chi_2).$$

**Proof.** Firstly, using the properties of Gauss sums, we get

$$\sum_{a=1}^{p-1}\overline{\psi}(a(a+1)) = \frac{1}{\tau(\psi)}\sum_{b=1}^{p-1}\psi(b)\sum_{a=1}^{p-1}\overline{\psi}(a)e\left(\frac{b(a+1)}{p}\right)$$

$$= \frac{\tau^2(\overline{\psi})}{\tau(\psi)} = \frac{\tau^3(\overline{\psi})}{p}. \tag{2}$$

On the other side, we get the sums

$$\sum_{a=1}^{p-1}\overline{\psi}(a(a+1)) = \psi(4)\sum_{a=0}^{p-1}\overline{\psi}\left(4a^2 + 4a\right)$$

$$= \psi(4)\sum_{a=0}^{p-1}\overline{\psi}\left((2a+1)^2 - 1\right) = \psi(4)\sum_{a=0}^{p-1}\overline{\psi}\left(a^2 - 1\right)$$

$$= \frac{\psi(4)}{\tau(\psi)}\sum_{b=1}^{p-1}\psi(b)\sum_{a=0}^{p-1}e\left(\frac{b(a^2-1)}{p}\right) = \frac{\psi(4)}{\tau(\psi)}\sum_{b=1}^{p-1}\psi(b)e\left(\frac{-b}{p}\right)\sum_{a=0}^{p-1}e\left(\frac{ba^2}{p}\right) \tag{3}$$

$$= \frac{\psi(4)\tau(\chi_2)}{\tau(\psi)}\sum_{b=1}^{p-1}\psi(b)\chi_2(b)e\left(\frac{-b}{p}\right) = \frac{\psi(4)\chi_2(-1)\tau(\chi_2)\tau(\psi\chi_2)}{\tau(\psi)}.$$

Combining Equations (2) and (3), we obtain

$$\tau^2(\overline{\psi}) = \left(\frac{-1}{p}\right)\psi(4)\tau(\chi_2)\tau(\psi\chi_2).$$

Now, Lemma 2 has been proved. □

**Lemma 3.** *Let $p$ be a prime with $p \equiv 1 \bmod 6$, $\chi$ be any sixth-order character* mod $p$. *Then, about classical Gauss sums $\tau(\chi)$, the following holds:*

$$\tau^3(\chi) + \tau^3(\overline{\chi}) = \begin{cases} p^{\frac{1}{2}}\left(d^2 - 2p\right) & \text{if } p = 12h + 1, \\ -i \cdot p^{\frac{1}{2}}\left(d^2 - 2p\right) & \text{if } p = 12h + 7, \end{cases}$$

*where $i^2 = -1$, $d$ is uniquely determined by $4p = d^2 + 27b^2$ and $d \equiv 1 \bmod 3$.*

**Proof.** Since $p \equiv 1 \bmod 6$, $\psi$ is a third-order character mod $p$. Any sixth-order character $\chi$ mod $p$ can be denoted as $\chi = \psi\chi_2$ or $\chi = \overline{\psi}\chi_2$. Note that $\psi^3(4) = 1$, $\overline{\psi}^3(4) = 1$ and $\chi_2^3 = \chi_2$, from Lemma 2 we deduce

$$\tau^6(\overline{\psi}) = \left(\frac{-1}{p}\right)\tau^3(\chi_2)\tau^3(\psi\chi_2) \tag{4}$$

and

$$\tau^6\left(\psi\right) = \left(\frac{-1}{p}\right)\tau^3(\chi_2)\tau^3\left(\overline{\psi}\chi_2\right).\tag{5}$$

Adding Equations (4) and (5), and then applying Lemma 1 we have

$$\left(\tfrac{-1}{p}\right)\tau^3(\chi_2)\left(\tau^3(\psi\chi_2) + \tau^3\left(\overline{\psi}\chi_2\right)\right) = \tau^6\left(\overline{\psi}\right) + \tau^6\left(\psi\right)\tag{6}$$
$$= \left(\tau^3\left(\overline{\psi}\right) + \tau^3\left(\psi\right)\right)^2 - 2p^3 = d^2p^2 - 2p^3.$$

Note that $\chi_2$ is a real character mod $p$, $\overline{\psi}\chi_2 = \overline{\psi\chi_2}$, and $\tau(\chi_2) = \sqrt{p}$. If $p \equiv 1 \bmod 4$; $\tau(\chi_2) = i \cdot \sqrt{p}$, $i^2 = -1$, if $p \equiv 3 \bmod 4$. From Equation (6) we may immediately prove the sum

$$\tau^3(\psi\chi_2) + \tau^3\left(\overline{\psi}\chi_2\right) = \begin{cases} p^{\frac{1}{2}}\left(d^2 - 2p\right) & \text{if } p = 12h+1, \\ -i\cdot p^{\frac{1}{2}}\left(d^2 - 2p\right) & \text{if } p = 12h+7. \end{cases}\tag{7}$$

Let $\chi = \psi\chi_2$, then $\chi$ is a sixth-order character mod $p$ and $\overline{\psi}\chi_2 = \overline{\chi}$. From Equation (7) we can deduce the sum term

$$\tau^3(\chi) + \tau^3\left(\overline{\chi}\right) = \begin{cases} p^{\frac{1}{2}}\left(d^2 - 2p\right) & \text{if } p = 12h+1, \\ -i\cdot p^{\frac{1}{2}}\left(d^2 - 2p\right) & \text{if } p = 12h+7. \end{cases}$$

The proof of Lemma 3 has been completed. $\square$

**Lemma 4.** *Let $p$ be a prime with $p \equiv 7 \bmod 12$, $\psi$ be any three-order character* mod $p$. *Then, we compute the sum term*

$$\frac{\tau^3\left(\overline{\psi}\right)}{\tau^3\left(\psi\right)} + \frac{\tau^3\left(\psi\right)}{\tau^3\left(\overline{\psi}\right)} = \frac{i\cdot\left(2p - d^2\right)}{p}.$$

**Proof.** Let $\psi$ be a three-order character mod $p$. Then, for any six-order character $\chi$ mod $p$, we must have $\chi = \psi\chi_2$ or $\chi = \overline{\chi}\chi_2$. Without loss of generality we suppose that $\chi = \psi\chi_2$, then note that $\psi(-1) = 1$, $\chi_2(-1) = -1$ and Theorem 7.5.4 in Reference [10], we acquire

$$\sum_{a=0}^{p-1} e\left(\frac{ba^2}{p}\right) = \chi_2(b)\cdot\sqrt{p},\ (p,b) = 1.$$

Using the properties of Gauss sums we can write

$$\sum_{a=0}^{p-1}\chi\left(a^2 - 1\right) = \frac{1}{\tau(\overline{\chi})}\sum_{b=1}^{p-1}\overline{\chi}(b)\sum_{a=0}^{p-1} e\left(\frac{b(a^2-1)}{p}\right)$$
$$= \frac{1}{\tau(\overline{\chi})}\sum_{b=1}^{p-1}\overline{\chi}(b)e\left(\frac{-b}{p}\right)\sum_{a=0}^{p-1} e\left(\frac{ba^2}{p}\right) = \frac{\sqrt{p}}{\tau(\overline{\chi})}\sum_{b=1}^{p-1}\overline{\chi}(b)\chi_2(b)e\left(\frac{-b}{p}\right)\tag{8}$$
$$= \frac{\overline{\chi}(-1)\chi_2(-1)\sqrt{p}\,\tau(\overline{\chi}\chi_2)}{\tau(\overline{\chi})} = \frac{\sqrt{p}\,\tau(\overline{\chi}\chi_2)}{\tau(\overline{\chi})}.$$

Noting that $\overline{\chi}^2 = \overline{\psi}^2 = \psi$, we can deduce

$$\sum_{a=0}^{p-1}\chi\left(a^2 - 1\right) = \sum_{a=0}^{p-1}\chi\left((a+1)^2 - 1\right) = \sum_{a=1}^{p-1}\chi(a)\chi(a+2)$$
$$= \frac{1}{\tau(\overline{\chi})}\sum_{b=1}^{p-1}\overline{\chi}(b)\sum_{a=1}^{p-1}\chi(a)e\left(\frac{b(a+2)}{p}\right) = \frac{\tau(\chi)}{\tau(\overline{\chi})}\sum_{b=1}^{p-1}\overline{\chi}^2(b)e\left(\frac{2b}{p}\right)\tag{9}$$
$$= \frac{\overline{\psi}(2)\tau(\chi)\tau(\psi)}{\tau(\overline{\chi})}.$$

Obviously, $\overline{\chi}\chi_2 = \overline{\psi}$ and $\psi^3(2) = 1$, applying Equations (8) and (9) we have

$$\tau^3(\chi) = p^{\frac{3}{2}} \cdot \frac{\tau^3\left(\overline{\psi}\right)}{\tau^3\left(\psi\right)}. \tag{10}$$

Similarly, we can see

$$\tau^3\left(\overline{\chi}\right) = p^{\frac{3}{2}} \cdot \frac{\tau^3\left(\psi\right)}{\tau^3\left(\overline{\psi}\right)}. \tag{11}$$

Combining Equation (10), Equation (11) and Lemma 3 we compute

$$\frac{\tau^3\left(\psi\right)}{\tau^3\left(\overline{\psi}\right)} + \frac{\tau^3\left(\overline{\psi}\right)}{\tau^3\left(\psi\right)} = \frac{1}{p^{\frac{3}{2}}}\left(\tau^3(\chi) + \tau^3\left(\overline{\chi}\right)\right) = \frac{i \cdot (2p - d^2)}{p}.$$

This completes the proof of Lemma 4.  □

**Lemma 5.** *Let $p$ be a prime with $p \equiv 1$ mod 12, $\psi$ be any three-order character mod $p$. Then, we obtain the sum term*

$$\frac{\tau^3\left(\overline{\psi}\right)}{\tau^3\left(\psi\right)} + \frac{\tau^3\left(\psi\right)}{\tau^3\left(\overline{\psi}\right)} = \frac{d^2 - 2p}{p}.$$

**Proof.** From Lemma 3 and the method of proving Lemma 4 we can easily deduce Lemma 5.  □

### 3. Proofs of the Theorems

In this section, we prove our two theorems. For Theorem 1, since $p \equiv 1$ mod 12, $\psi$ is a third-order character mod $p$, then for any positive integer $k$, let

$$U_k(p) = \frac{\tau^{3k}\left(\psi\right)}{\tau^{3k}\left(\overline{\psi}\right)} + \frac{\tau^{3k}\left(\overline{\psi}\right)}{\tau^{3k}\left(\psi\right)}.$$

From Lemma 5 we have

$$U_1(p) = \frac{\tau^3\left(\overline{\psi}\right)}{\tau^3\left(\psi\right)} + \frac{\tau^3\left(\psi\right)}{\tau^3\left(\overline{\psi}\right)} = \frac{d^2 - 2p}{p} \tag{12}$$

and

$$\frac{d^2-2p}{p} \cdot U_k(p) = U_k(p)U_1(p) = \left(\frac{\tau^{3k}(\psi)}{\tau^{3k}\left(\overline{\psi}\right)} + \frac{\tau^{3k}\left(\overline{\psi}\right)}{\tau^{3k}(\psi)}\right) \cdot \left(\frac{\tau^3\left(\overline{\psi}\right)}{\tau^3(\psi)} + \frac{\tau^3(\psi)}{\tau^3\left(\overline{\psi}\right)}\right)$$

$$= \frac{\tau^{3k+3}\left(\overline{\psi}\right)}{\tau^{3k+3}(\psi)} + \frac{\tau^{3k+3}(\psi)}{\tau^{3k+3}\left(\overline{\psi}\right)} + \frac{\tau^{3k-3}\left(\overline{\psi}\right)}{\tau^{3k-3}(\psi)} + \frac{\tau^{3k-3}(\psi)}{\tau^{3k-3}\left(\overline{\psi}\right)} = U_{k+1}(p) + U_{k-1}(p). \tag{13}$$

Combining Equations (12) and (13) we may immediately compute the second-order linear recursive formula

$$U_{k+1}(p) = \frac{d^2 - 2p}{p} \cdot U_k(p) - U_{k-1}(p) \tag{14}$$

with initial values $U_0(p) = 2$ and $U_1(p) = \frac{d^2-2p}{p}$.

Note that the two roots of the equation $\lambda^2 - \frac{d^2-2p}{p}\lambda + 1 = 0$ are

$$\lambda_1 = \frac{d^2 - 2p + 3dbi\sqrt{3}}{2p} \quad \text{and} \quad \lambda_2 = \frac{d^2 - 2p - 3dbi\sqrt{3}}{2p}.$$

So from Equation (14) and its initial values we may immediately deduce the general term

$$U_k(p, \psi) = \left( \frac{d^2 - 2p + 3dbi\sqrt{3}}{2p} \right)^k + \left( \frac{d^2 - 2p - 3dbi\sqrt{3}}{2p} \right)^k,$$

where $i^2 = -1$. Now Theorem 1 has been finished.

Similarly, from Lemma 4 and the method of proving Theorem 1 we can easily obtain Theorem 2. Now, we have completed all the proofs of our Theorems.

## 4. Conclusions

The main results of this paper are Theorem 1 and 2. They give a new second-order linear recurrence formula for Equation (1) with the third-order character $\psi$ mod $p$. Therefore, we can calculate the exact value of Equation (1). Note that $|\tau(\overline{\psi})/\tau(\psi)| = 1$, so $\tau(\overline{\psi})/\tau(\psi)$ is a unit root, thus, the results in this paper profoundly reveal the distributional properties of two different Gauss sums quotients on the unit circle.

For the other mod $p$ characters, for example, the fifth-order character $\chi$ mod $p$ with $p \equiv 1$ mod 5, we naturally ask whether there exists a similar formula as presented in our theorems. This is still an open problem. It will be the content of our future investigations.

**Funding:** This research was funded by [National Natural Science Foundation of China] Grant number [11771351].

**Acknowledgments:** The author wish to express her gratitude to the editors and the reviewers for their helpful comments.

**Conflicts of Interest:** The authors declare no conflict of interest.

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
