# Peer review of "On Classical Gauss Sums and Some of Their Properties"

_symmetry, doi:10.3390/sym10110625_

Reviewer 1 Report

On page 2 lines 4-5, I don't understand  the sentence "But as we all know that ---in Gauss sums." The author should correct it in appropriate way.

Page 3, middle, "elementary number theory and analytic number theory" -> elementary and analytic number theory.

I recommend the acceptance of the present paper on condition that the above comments are

incorporated in the revised version of the paper.

Author Response

Dear Editor :

Thank you very much for your report on my manuscript (ID: symmetry-382733) titled “On

the classical Gauss sums and their some recursive properties” by Li Chen. I think the reviewers

’ comments are very important. According to the suggestions, I have made some improvements

on my manuscript, including the following aspects:

Point 1: On page 2 lines 4-5, I don't understand  the sentence "But as we all know that ---in Gauss sums." The author should correct it in appropriate way.

Response 1: I have revised the old sentence to: However, perhaps the most beautiful and important property about Gauss sums τ(χ) is that |τ(χ)| =√q, for any primitive character χ mod q.And I marked them in red highlighting in PDF.

Point 2: Page 3, middle, "elementary number theory and analytic number theory" -> elementary and analytic number theory.

Response 2: I have changed them and I marked them in red highlighting in PDF.

After these modifications, I think the new version of my paper is much better than its

before. Now, I submit the revised version of our paper to your journal, and hope it can be

published in your nice journal. Thank you!

Sincerely yours

Li Chen

Reviewer 2 Report

1.This is a short study on some properties of Gauss sums.

2.Finding properties of Gauss sums is important in Analysis.A few nontrivial and useful properties have been found. Theorem 1 is shown nicely and it the best result in the paper.

3.I do not have any major recommendation for the improvement of this paper,since the paper is written properly. 4.A better title is "On the classical Gauss sums and some of their properties

Author Response

Dear editor:

     Thank you so much for your approval of the article. I have changed my  title to "On the classical Gauss sums and some of their properties". And I marked them in yellow highlighting in the PDF file I uploaded.

 yours' sincerely

Li Chen

Reviewer 3 Report

the article is well written, the abstract and the introduction are clear; however in the introduction is mentioned the Legendre symbol and some related properties, it then could be useful to cite related paper.

the section 2. is well written, in the third section it could better to make a brief explanation of the results obtained, and moreover it is necessary to conclude the papers with adequate conclusions.

Author Response

Dear Editor :

Thank you very much for your report on my manuscript (ID: symmetry-382733) titled “On

the classical Gauss sums and their some recursive properties” by Li Chen. I think the reviewers

’ comments are very important. According to the suggestions, I have made some improvements

on my manuscript, including the following aspects:

Point 1: however in the introduction is mentioned the Legendre symbol and some related properties, it then could be useful to cite related paper.

Response 1: I have added the explanation in blue highlighting in the PDF.

Point 2: in the third section it could better to make a brief explanation of the results obtained, and moreover it is necessary to conclude the papers with adequate conclusions.

Response 2: I have added the fourth part in my paper, it has adequate explanation. And I marked it in green highlighting in the PDF.

After these modifications, I think the new version of my paper is much better than its

before. Now, I submit the revised version of our paper to your journal, and hope it can be

published in your nice journal. Thank you!

Sincerely yours

Li Chen